# Specialized areas for value updating and goal selection in the primate orbitofrontal cortex

**Elisabeth A Murray\*, Emily J Moylan, Kadharbatcha S Saleem, Benjamin M Basile, Janita Turchi**

Section on the Neurobiology of Learning and Memory, Laboratory of Neuropsychology, National Institute of Mental Health, National Institutes of Health, Bethesda, United States

**Abstract** The macaque orbitofrontal cortex (OFC) is essential for selecting goals based on current, updated values of expected reward outcomes. As monkeys consume a given type of reward to satiety, its value diminishes, and OFC damage impairs the ability to shift goal choices away from devalued outcomes. To examine the contributions of OFC's components to goal selection, we reversibly inactivated either its anterior (area 11) or posterior (area 13) parts. We found that neurons in area 13 must be active during the selective satiation procedure to enable the updating of outcome valuations. After this updating has occurred, however, area 13 is not needed to select goals based on this knowledge. In contrast, neurons in area 11 do not need to be active during the value-updating process. Instead, inactivation of this area during choices causes an impairment. These findings demonstrate selective and complementary specializations within the OFC.

**\*For correspondence:** murraye@
mail.nih.gov

**Competing interests:** The authors declare that no competing interests exist.

## Introduction

Damage to the prefrontal cortex often disrupts the translation of knowledge into action. Patients with such lesions can accurately explain various intentions, rules and social conventions, but nevertheless violate them (*Milner, 1963*; *Luria, 1966*). As *Teuber (1972a)* put it: "Patients with frontal lobe disease seem to perceive the mistakes they make, but are unable to use the information to guide their behavior."

As a result, it has been argued that a principal function of the frontal lobe lies in translating knowledge into behavioral goals (*Teuber, 1972b*). The uncoupling of knowledge from goals has been generalized as *goal neglect* (*Duncan et al., 1996*; *2008*), and this concept extends beyond laboratory tasks, such as card sorting, to the social domain. For example, patients with damage to the orbital and medial prefrontal cortex can accurately convey appropriate moral principles and social conventions, but then violate them in the course of their behavior (*Bechara et al., 1994*). In some cases, this deficit occurs in isolation, with normal behavior in other cognitive domains (*Damasio et al., 1991*; *Fellows, 2011*). An impairment in translating knowledge into behavioral goals could be fundamental to many addictive behaviors, compulsive disorders, and psychopathologies.

A simple account for goal neglect is that the cognitive processes leading to goal selection have, in such cases, poor access to knowledge about expected outcomes and their value. One way to assess an animal's knowledge of outcome values is through reinforcer devaluation in the *devaluation task*. Although the subjective value of a food or fluid outcome is influenced by many factors, such as probability, magnitude, and the effort required to obtain it, the devaluation task isolates a different

**eLife digest** Everyone knows that somehow, somewhere, the brain translates knowledge into action. In some people, however, knowledge and action become disconnected. These people behave in a way that either ignores or contradicts the knowledge that they have. They know what to do and can explain it to others, but – when the time comes to act – they do something else, something wrong.

Murray et al. have now investigated how a brain region called the orbitofrontal cortex helps to link knowledge and action in macaque monkeys, which, unlike rodents, have all of the main brain areas that make up the orbitofrontal cortex of humans. The monkeys learned to associate images with different types of food, and then performed a task where they chose between two images in order to get the food they wanted. On some days, one of the foods was less 'valuable' because the monkeys had already eaten a lot of it. In these circumstances, monkeys chose fewer of the images associated with that food.

By temporarily inactivating either the front or back region of the monkey's orbitofrontal cortex at different times, Murray et al. showed that these regions make different contributions to decision making. Inactivating the back region of the orbitofrontal cortex disrupted the ability of monkeys to update their knowledge about the value of a particular foodstuff. However, inactivating the front part of the orbitofrontal cortex disrupted their ability to use this knowledge to select the images that led to the most valuable food. This contradicts the widely held belief that the orbitofrontal cortex acts as a single entity to update values and translate this knowledge into action.

Future work will need to investigate how, having translated knowledge into a chosen action, the orbitofrontal cortex stimulates the motor areas of the brain to generate the movements needed to perform that action.

and independent aspect of subjective value: an outcome's worth at a particular moment based on the individual's current state. In one version of this task, monkeys first learn that some objects are associated with one kind of food (food 1), while other objects are associated with a different food (food 2). Next, a selective satiation procedure temporarily devalues one food type, and monkeys are given a series of choice tests in which food-1 objects are pitted against food-2 objects. If monkeys can update the value of expected outcomes and link this information to their goal choices, they will shift these choices away from objects associated with the devalued food, a phenomenon called the *devaluation effect*. Other kinds of visual stimuli can be substituted for physical objects in these experiments, as in the present experiment.

The key question for the present study is *when* various brain areas make their contribution to the devaluation effect. By inactivating the amygdala of monkeys either before or after the selective satiation procedure, *Wellman et al. (2005)* showed that neurons in the amygdala need to be active during the satiation procedure for normal value updating to occur. Specifically, inactivation of the amygdala before selective satiation disrupted devaluation effects, whereas inactivation after satiation, but before the choice phase of the experiment, had no effect. Thus, the amygdala is essential for value updating but, once that has occurred, it is no longer essential for making goal choices based on those valuations.

In addition to the amygdala, the orbitofrontal cortex (OFC) makes a necessary contribution to devaluation effects in monkeys (*Izquierdo et al., 2004*; *Machado and Bachevalier, 2007*; *Baxter et al., 2009*; *Rudebeck et al., 2013*). Further, functional interaction of the amygdala and OFC is required (*Baxter et al., 2000*). The granular prefrontal areas that compose the primate OFC are also essential for representing expected outcomes more generally, including both their sensory properties and subjective value (*Padoa-Schioppa, 2011*; *Rudebeck and Murray, 2014*), and the agranular OFC areas in rodents have similar properties (*Schoenbaum et al., 2009*). Within the primate OFC, areas 13 and 11, considered together, are necessary for normal devaluation effects, but area 14 is not (*Rudebeck and Murray, 2011*). Thus, the functional contribution of the OFC to this kind of value-based goal selection is mediated by the *sensory network* proposed by *Carmichael and Price (1996),* which receives inputs from gustatory, olfactory, visceral, and visual sensory areas (*Carmichael and Price, 1995*; *Saleem et al., 2008*). Each of these inputs presumably contributes to

representations of outcome expectancies. Consistent with this idea, when monkeys view images that predict specific amounts or types of outcomes, the activity of neurons in both areas 13 and 11 signals the expected value of those upcoming rewards (*Hikosaka and Watanabe, 2000*; *Tremblay and Schultz, 2000*; *Wallis and Miller, 2003*; *Padoa-Schioppa and Assad, 2006*).

Area 11 lies anterior to area 13, and these areas differ in both cytoarchitecture and connections (*Preuss and Goldman-Rakic, 1991*; *Carmichael and Price, 1994*; *Saleem et al., 2008*). Little, however, is known about their functional specializations. Anterior–posterior dissociations have been reported in electrophysiological studies in macaques (*Mora et al., 1980*; *Kobayashi et al., 2010*), and in functional neuroimaging studies in humans (*Sescousse et al., 2010*; *Klein-Flugge et al., 2013*), but these correlational findings leave open the causal contributions of these two OFC components to behavior. Here we show that they make selective and complementary contributions to reward-based goal selection, and in particular to translating abstract, updated knowledge about outcome valuations into advantageous choices.

## Results

Monkeys were trained on a series of visual discrimination problems (*Figure 1a*), which provided the opportunity to learn about a large number of images and their associated food rewards. This set the stage for later probe tests (*Figure 1b*), which pitted food-1 images against food-2 images in a series of choice trials. Importantly, different pairs of images appeared on each trial. To test whether the OFC contributes to different components of the devaluation effect, we infused the GABA$_A$ agonist 4,5,6,7-tetrahydroisoxazolo [5,4-c]pyridin-3-ol hydrochloride (THIP hydrochloride, hereafter THIP) into area 13 or area 11 before and after selective satiation. Infusions administered before the selective satiation procedure inactivated the OFC *during* the selective satiation procedure as well as during the choice trials of the probe test. Thus, disruption of devaluation effects by THIP infusions before selective satiation would indicate a failure of either value updating or goal selection. Infusion of THIP after the selective satiation procedure inactivated the OFC only during the goal selection in the choice phase of the experiment. Thus, disruption of devaluation effects by THIP infusions after selective satiation would indicate a selective contribution to goal selection. (Note that this use of the term *goal* differs from its use in terms of *goal-directed behavior*, in which the goal corresponds to a reward outcome.)

Infusions targeted area 13 and area 11 separately (*Figure 1c*). Monkeys were trained on the main task and then received surgery to implant a chamber on the cranium (see Materials and methods). Structural MR scans were used to identify and localize OFC subsectors, and additional structural scans combined with infusion of a gadolinium-saline solution were used to confirm our ability to infuse drugs precisely into the target sites before beginning data collection (*Figure 2a*). In addition, at the conclusion of the experiment, some monkeys received retrograde tracer injections at the same sites targeted by the drug infusions (*Figure 2b–d*). This allowed us to verify that the infusion sites identified by structural MR scans at the beginning of the experiment were as intended throughout the study.

We used a within-subjects design. Each of five monkeys received infusions of THIP or saline within area 13 and, in separate sessions, area 11, both before and after selective satiation. Sites in areas 13 and 11 were roughly 5 mm apart in the anteroposterior dimension (*Figure 1c,d*). To collect these data, the reinforcer devaluation was conducted repeatedly, with a minimum of one week between selective satiation procedures. Image choices from probe tests run after reinforcer devaluation (Day 4) were compared to baseline choices obtained that same week (Day 2, *Figure 3*); a proportion shifted score was calculated by averaging scores for each pair of probe tests for each condition (see Materials and methods).

### Effects of reversible inactivations

As expected, when saline was infused into either area 13 or 11 the monkeys showed robust devaluation effects (*Figure 4*). That is, after selective satiation, monkeys shifted their choices to images that yielded the higher value outcome. The effect of temporary inactivations of the OFC varied markedly across conditions. A repeated-measures ANOVA on proportion shifted with factors of Treatment (THIP, saline), Region (area 13, area 11), and Time (before satiation, after satiation) revealed a significant three-way interaction ($F_{(1,4)} = 16.82$, p = 0.015, partial $\eta^2 = 0.81$), indicating that behavioral

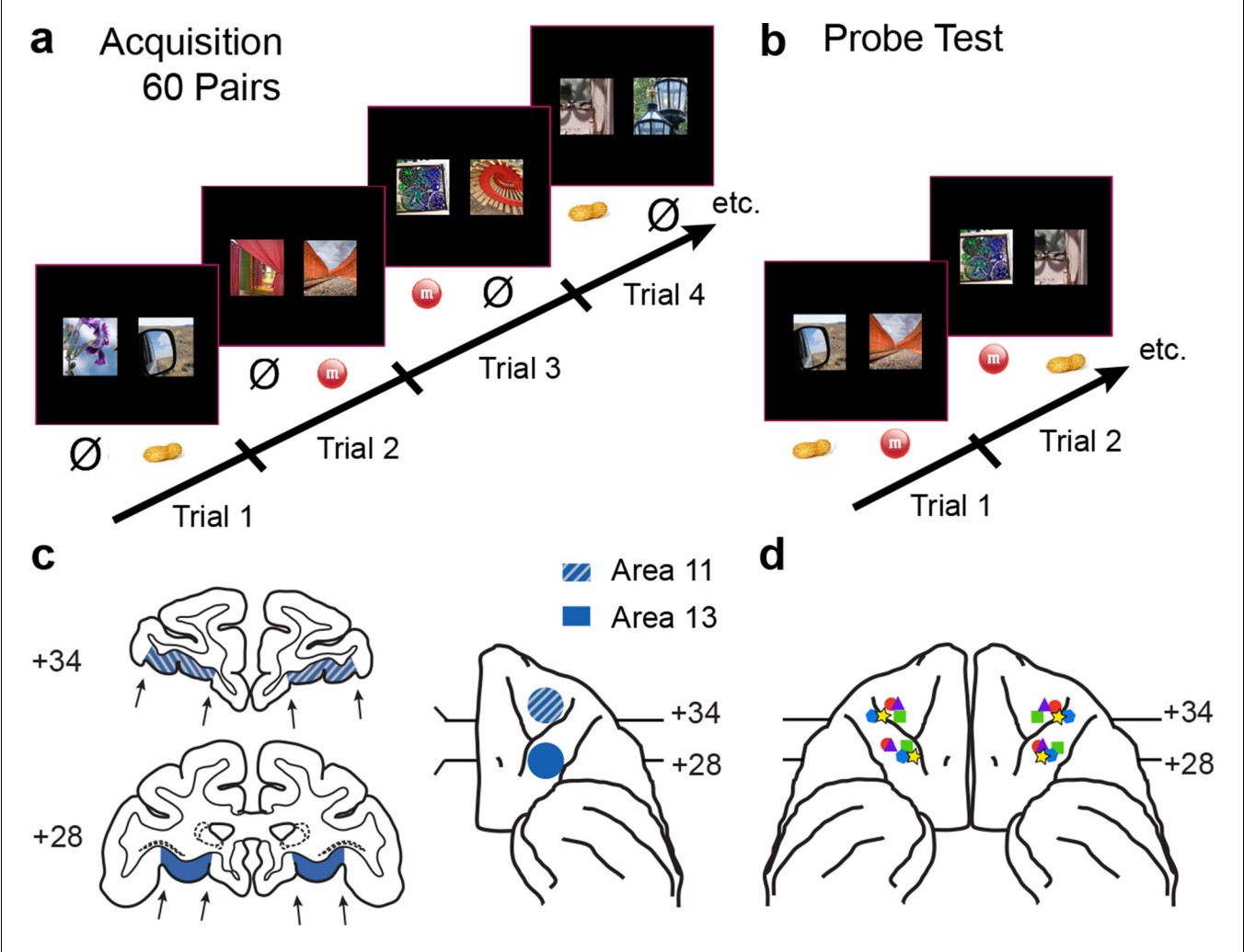

**Figure 1.** Behavioral task and localization of infusions. (**a**) 60-pair discrimination learning task. On each trial, monkeys chose one of two images by making a touch to the screen. Touching the image designated the S+ (rewarded) led to delivery of food reward, whereas touching the image designated the S− (nonrewarded) led to termination of the trial and initiation of the intertrial interval. Half of the S+ led to delivery of food 1 (e.g., peanut); the remaining S+ led to delivery of food 2 (e.g., M&M). Image pairs and image-reward assignments were fixed across sessions. There were 60 trials per session. (**b**) Probe tests. On each trial, monkeys chose between a food-1 and food-2 associated image. Touching the image on the screen led to delivery of the food reward assigned to that image in the acquisition phase. Images were paired anew each session. There were 30 trials per session. (**c**) Intended target locations. Left side: coronal sections through the frontal lobe. Right side: intended target locations illustrated on a ventral view of the anterior half of the macaque brain. Numerals indicate distance in mm from the interaural plane (0). (**d**) Reconstruction of infusion sites onto ventral view of the macaque brain. Symbols indicate location of infusion sites; different symbols correspond to individual monkeys. For ease in viewing, left hemisphere sites are shown on the left.

effects of inactivation differed both as a function of area and of timing. Post-hoc tests, detailed below, revealed a double dissociation of function within the OFC. When monkeys selected goals based on updated subjective valuations, inactivation of area 13 but not area 11 during the value-updating phase caused an impairment. In contrast, inactivation of area 11 but not area 13 during the goal-selection phase of the experiment caused an impairment.

Scores obtained when saline was infused before selective satiation did not differ from those obtained when saline was infused after selective satiation (repeated-measures ANOVA with factors of Region and Time; for all main effects and interactions: $F_{(1,4)} < 1.48$, $p > 0.29$); we therefore pooled saline data within each region for subsequent analysis. Planned follow-up comparisons revealed that THIP infusions into area 13 before satiation significantly lowered proportion shifted scores relative to those obtained with saline infusions (two-tailed paired t-tests with Bonferroni-corrected alpha

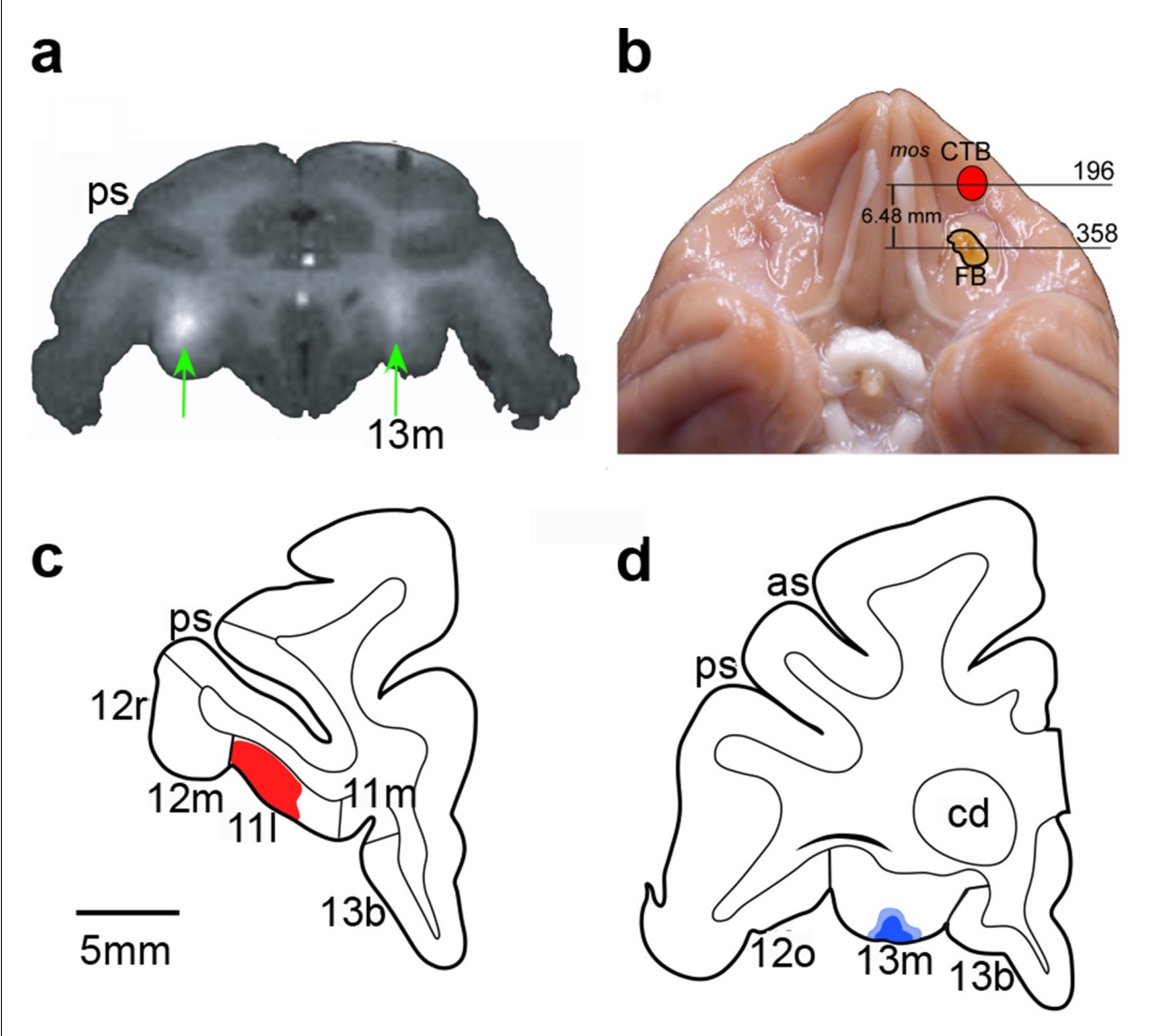

**Figure 2.** Documentation of infusion sites in the OFC. (a) MR image showing sites of gadolinium-saline infusions (white hypersignal) in area 13. A gadolinium-saline solution was infused to a site just dorsal to the intended target to confirm accurate placement of the cannulae before data collection began. (b) Photograph of ventral view of brain showing locations of retrograde tracer injections made at the end of the experiment to reconfirm target locations. Numerals correspond to sections illustrated in (c) and (d). Red outlined region illustrates site of injection of cholera toxin subunit B (CTB), as reconstructed after histological processing of tissue. Yellow outlined region illustrates site of injection of Fast Blue, which appears yellow before processing. (c) and (d). Line drawings of coronal sections through the anterior OFC (area 11) (c) and the posterior OFC (area 13) (d). Red-shaded region corresponds to CTB tracer injection. Blue-shaded region corresponds to Fast Blue injection. Scale bar applies to both c and d. Compare and contrast with *Figure 1*. Abbreviations: cd, caudate nucleus; ps, principal sulcus; as, arcuate sulcus; 11l, 11m, 12m, 12r, 12o, 13m, and 13b, cytoarchitectonic divisions of the prefrontal cortex (*Carmichael and Price, 1994*).

= 0.025 for all follow-up tests; $t_4$ = 5.61, p = 0.005, d = 2.51). Infusions of THIP into area 13 after satiation had no effect ($t_4$ = 0.24, p = 0.826). Because normal functioning of area 13 was necessary during, but not after, satiation, area 13 appears to be essential for value updating but not for selecting visual goals based on that information (*Figure 4*, left). The converse pattern of results was obtained for area 11. THIP infusions into area 11 before satiation had no effect on behavior ($t_4$ = 0.44, p = 0.686), whereas THIP infusions after satiation disrupted devaluation effects ($t_4$ = 4.36, p = 0.012, d = 1.95). Thus, area 11 appears to be essential for advantageous goal selection but not value updating (*Figure 4*, right). We also conducted the analysis using proportion choices of images

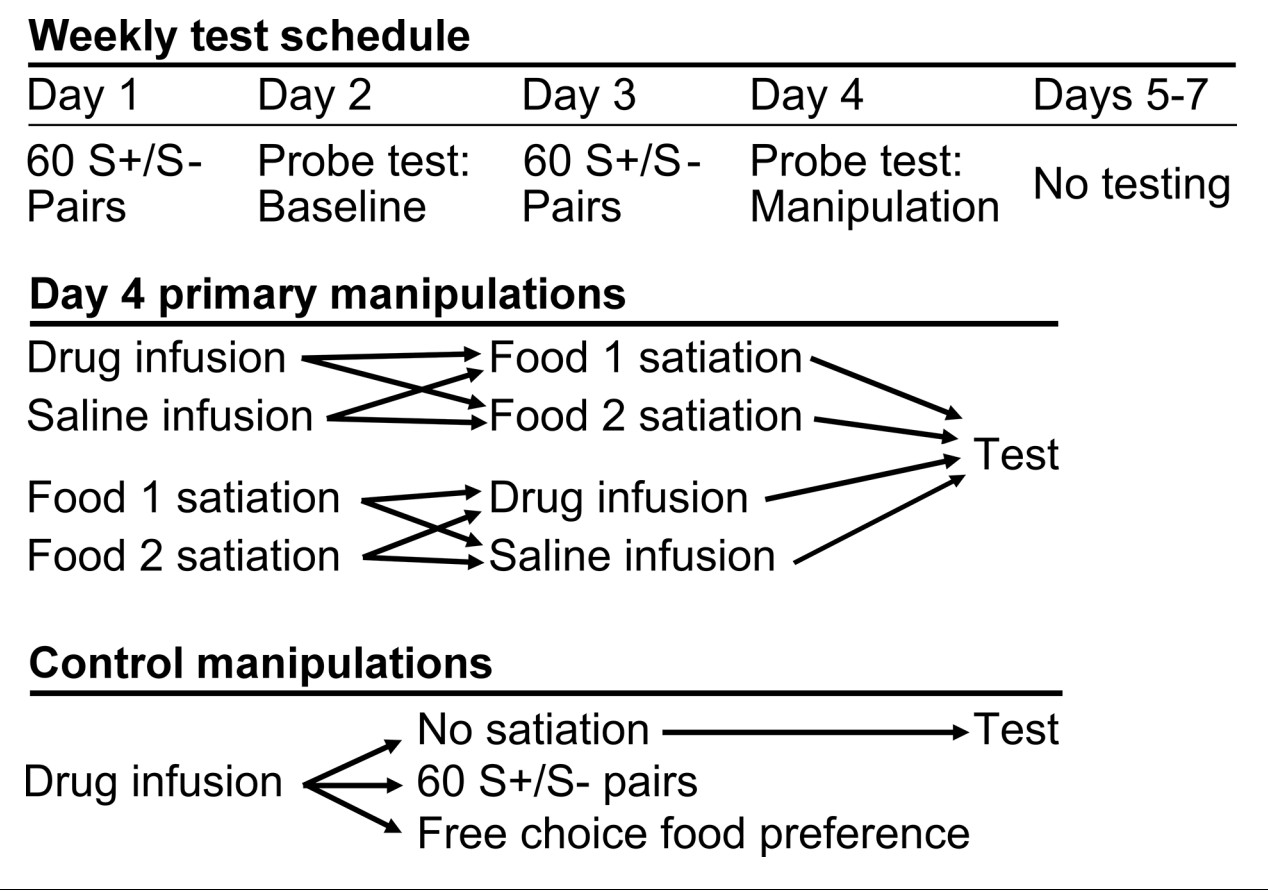

**Figure 3.** Schematic of experimental design. During the main experiment, monkeys were tested 4 days per week. All primary experimental manipulations were carried out on Day 4.

associated with the nonsated food as the dependent measure, and found the same pattern of results (area 13, pre-sate vs. saline: $t_4 = 5.95$, $p = 0.004$, $d = 2.66$, post-sate vs. saline: $t_4 = -0.14$, $p = 0.898$; area 11, pre-sate vs. saline: $t_4 = 0.46$, $p = 0.669$, post-sate vs. saline: $t_4 = 4.72$, $p = 0.009$, $d = 2.11$).

### Control procedures

We conducted three control procedures. First, we considered whether the infusions might disrupt monkeys' choices in the absence of selective satiation. To examine this possibility, in each of the five subjects we infused THIP into either area 11 or area 13 and administered probe tests without any prior selective satiation. These sessions were inserted randomly within the infusion series carried out for the main study. As shown in *Figure 5A*, in the absence of satiation, infusions of THIP into either area did not change monkeys' choices of images paired with each food type (repeated-measures ANOVA; $F_{(2,8)} = 0.06$, $p = 0.945$). These data, together with the lack of effect of THIP infusions either after selective satiation in area 13 or before selective satiation in area 11 (*Figure 4*), rules out the possibility that infusion of THIP, per se, disrupted monkeys' choices.

A more serious concern was that inactivation of area 11 might disrupt image choices generally, as opposed to disrupting image choices based on current value, as we assumed. Accordingly, we carried out a second control procedure in four of the original five monkeys, in which we tested the effect of THIP infusions into area 11 on performance of the 60 discrimination problems that had been learned in the training phase (*Figure 1a*). THIP infusions into area 11 before test sessions did not affect accuracy on the familiar 60 image pairs ($t_3 < 0.001$, $p > 0.999$; *Figure 5B*), response speed on correct trials (median response latencies, mean (SEM); no-drug baseline = 1.53 s (0.17); THIP

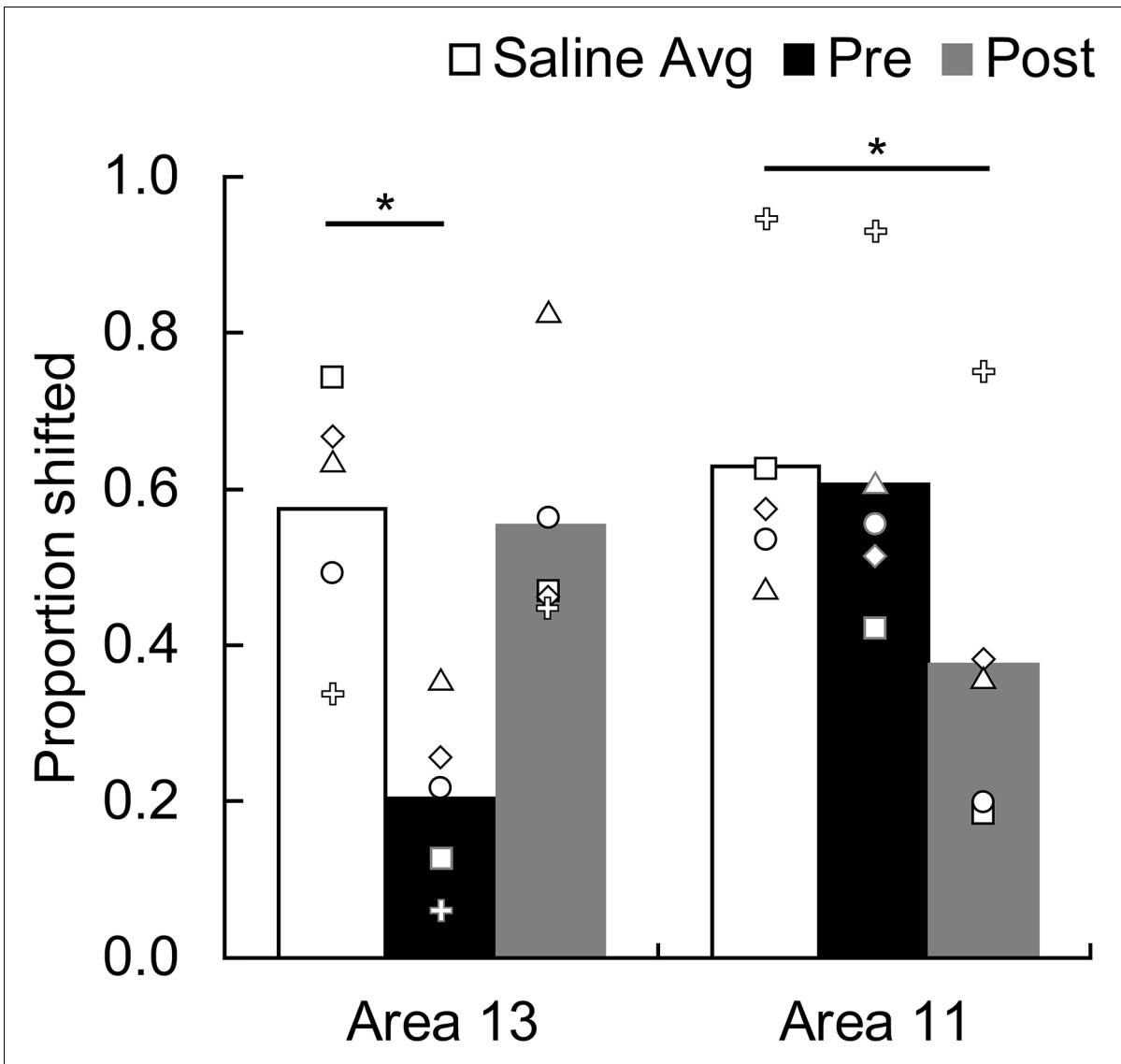

**Figure 4.** Effect of temporary inactivation of area 13 and area 11 on image choices following reinforcer devaluation. Proportion shifted represents the shift in image choices after selective satiation (Day 4) relative to baseline (Day 2), combined across probe tests; the higher the score the greater the shift away from choices associated with the devalued food. Asterisks indicate significant differences (area 13, saline vs. THIP Pre, p = 0.005; area 11, saline vs. THIP Post, p = 0.012). Bars represent group means and symbols show the scores of individual monkeys. White bars, saline infusions; black bars, THIP infusions administered before selective satiation; gray bars, THIP infusions administered after selective satiation.

infusion before = 1.35 s (0.17); $t_3$ = 1.45, p = 0.242), or time to complete test sessions (total session duration, mean (SD); no-drug baseline = 449.8 s (46.69); THIP infusion before = 468.1 s (54.92); $t_3$ = 1.63, p = 0.201). These data, together with the lack of effect of inactivation of area 11 in our first control procedure, described above, led us to conclude that THIP infusions into area 11 after selective satiation in our main task specifically disrupted retrieval or use of the recently updated value, as opposed to having a general effect on choosing rewarded images.

Finally, we considered whether infusions affected satiety mechanisms directly, in the absence of the need to select goals in the form of visual images that had been paired with food. This third control procedure was run after the data from the main experiment had been collected. In two subjects from the main experiment and a third monkey used for this procedure only, we assessed monkeys' food choices directly, in the absence of images, to determine if the selective satiation procedure had been effective. Monkeys received a series of 30 choice trials in which food 1 was directly pitted

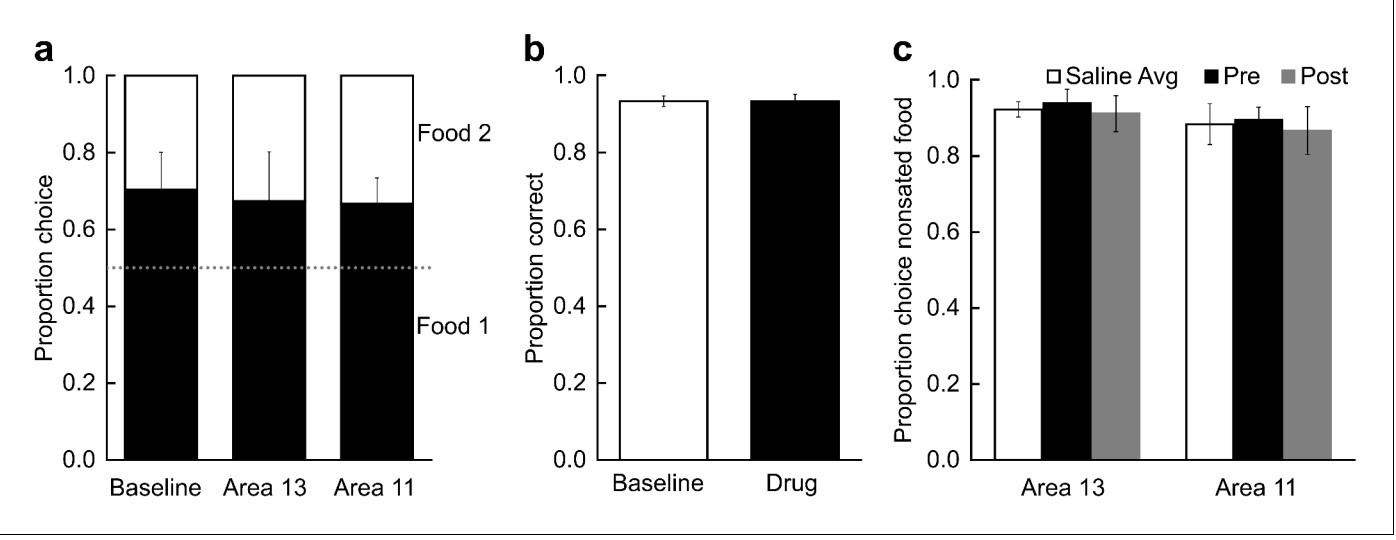

**Figure 5.** Control procedures. (**a**) Effect of temporary inactivation of area 13 and area 11 of the OFC on image choices in the absence of selective satiation. Proportion choice indicates the proportion of chosen images associated with the either the preferred food (black, designated Food 1) or nonpreferred food (white, designated Food 2) out of the total number of choices averaged across probe tests. There was no effect of THIP infusions on image choices relative to baseline. Baseline: image choices on baseline days; Area 13: image choices after THIP infusions into area 13; Area 11: image choices after THIP infusions into area 11. (**b**) Effect of THIP infusions into area 11 on image choices for the 60 object discrimination problems learned in the Training phase (*Figure 1a*). Proportion correct indicates the accuracy on the familiar discrimination problems. There was no effect of THIP infusions on choice accuracy for the familiar discrimination problems. Baseline, no infusions; Drug, THIP infusions into area 11 administered prior to test session. (**c**) Effect of temporary inactivation of area 13 and area 11 on food choices after selective satiation. Proportion choice nonsated food is scored from forced-choice trials involving selection between the sated and nonsated food. White bars, saline infusions; black bars, THIP infusions administered before selective satiation; gray bars, THIP infusions administered after selective satiation. All monkeys reliably chose the higher-value (nonsated) food after selective satiation procedures, even after inactivation of area 13 or area 11. Thus, satiety mechanisms were intact during all experimental conditions. For all panels, bars represent group means and error bars represent ± SEM.

against food 2. In separate sessions, saline and THIP were infused before and after selective satiation, and we recorded monkeys choices of food relative to baseline choices of food, just as we had for the probe tests with images. When monkeys could choose between visible foods directly, rather than choosing between images that had previously been paired with food, THIP infusions had no effect on choices, regardless of whether that was measured as a function of proportion shifted, as used for image choices, or as a function of the proportion choice of the nonsated food (repeated-measures ANOVA with factors of region and treatment; for all main effects of region: $F_{(1,2)} < 8.68$, $p > 0.098$; for all main effects of treatment and interactions: $F_{(2,4)} < 0.29$, $p > 0.764$; *Figure 5C*). Thus, consistent with our earlier work based on permanent lesions (*Izquierdo et al., 2004*; *Rudebeck et al., 2013*), the reversible inactivations had no discernable influence on satiety mechanisms, and the selective satiation procedure produced the desired effect.

## Discussion

### Goal neglect after inactivation of area 11

Our results reveal a double dissociation of function between the anterior and posterior components of the OFC, areas 11 and 13, which *Carmichael and Price (1996)* characterize as components of an orbitomedial prefrontal 'sensory network'. Their mutually supportive and complementary functions agree with this concept of prefrontal organization. When an outcome's value is altered, area 13—but not area 11—is essential for a value updating function. After updating has occurred, however, area 11—but not area 13—is essential for translating the knowledge about current valuations into advantageous choices. Thus, these two areas play complementary roles in mediating devaluation effects: the caudal part of the OFC updates valuations and the rostral part translates this knowledge into action.

The present experiment provides the first demonstration of a double dissociation of function between the anterior and posterior components of the primate OFC, which complements recent studies that have pointed to functional differences between its medial and lateral parts (*Bouret and Richmond, 2010*; *Noonan et al., 2010*; *Rudebeck and Murray, 2011*).

The finding of an impairment when area 11 was inactivated after—but not before—the selective satiation procedure is particularly significant. The monkeys had by all indications acquired information about the new, updated value of the food outcome, yet their choices did not reflect this. This observation shows that area 11 is not necessary for updating valuations per se, but rather for the use of this knowledge to choose goals based on visual stimuli associated with predicted food outcomes. In essence, once value updating has occurred, area 11 plays the dominant role and area 13 is no longer needed to guide choices between stimuli based on current valuations. Presumably, storage of the updated value occurred either during or shortly after the selective satiation procedure, a process that depended on area 13, and this information was then broadcast to other areas, including area 11. Inactivation of area 11 immediately before the probe tests prevented the optimal use of this knowledge.

An additional aspect of our double dissociation deserves comment. Inactivation of area 11 before the selective satiation procedure, which had no effect on behavior, would have persisted into the choice phase, when THIP infusions into area 11 did cause an impairment. But it is reasonable to expect an impairment to follow choice-phase inactivation of area 11, regardless of how long its neurons had been inactivated. We do not believe that the effect of pre-satiation THIP infusions could have dissipated in time for area 11 to participate in goal selection. Infusion of similar volumes of THIP into the pulvinar produced behavioral effects in monkeys lasting several hours (*Wilke et al., 2013*), and smaller volumes administered in rat superior colliculus affected behavior for at least two hours (*Di Scala et al., 1983*), both well beyond the elapsed time between pre-satiation infusion and choice tests in the present study. Instead, our negative results after pre-satiation infusions probably reflect the cooperative function of other prefrontal areas, which would also explain the relatively mild effect of choice-phase area 11 inactivations. The impairment when area 11 was inactive only during goal selection, but not when it was inactive during *both* value updating and goal selection, suggests that area 11 plays a dominant role in a partially-redundant goal-selection network. Another possibility is that when area 11 was inactivated before the selective satiation procedure, it was effectively taken 'offline', rendering it unable to receive updated valuation signals and therefore unable to participate in goal selection later. According to this account, when area 11 had received updated valuation signals during the satiation procedure it made a significant contribution to goal selection, but otherwise did not. Presumably, neighboring prefrontal areas could mediate goal selection in this circumstance.

## Independent contributions of anterior and posterior portions of the OFC

Notably, inactivation of area 13 yielded a pattern of results virtually identical to that observed after inactivation of basolateral amygdala in monkeys performing the same task (*Wellman et al., 2005*). Given the evidence from neuropsychological studies that the amygdala and the OFC must functionally interact in mediating devaluation effects (*Baxter et al., 2000*), the present results point to an interaction of the basolateral amygdala with area 13 to support value updating, specifically during the selective satiation procedure.

One implication of this result is that, during normal devaluation, values get updated gradually during the selective satiation procedure. This does not imply, however, that the areas involved in goal selection receive real-time information about this gradual devaluation. Instead, from a modeling perspective, the effect of satiation might be computed (read-in) at the time of the choice. These possibilities are not mutually exclusive, and competing choice-related areas might operate differently in this regard.

Area 13, specifically area 13m, is the first site of convergence in the OFC of gustatory, olfactory, somatic, visceral, and visual sensory inputs. On these grounds, it has been proposed to be the site of sensory representations of food outcomes, including the sensory properties of the food as well as its value (*Carmichael and Price, 1996*; *Critchley and Rolls, 1996*; *Rolls, 2000*; *Saleem et al., 2008*). Consistent with this idea, neurons in area 13 of macaques recorded during performance of reward-guided tasks signal the value and identity of expected outcomes (*Tremblay and Schultz, 1999*;

*Wallis and Miller, 2003*; *Padoa-Schioppa and Assad, 2006*; *2008*; *2009*), as do neurons in the amygdala (*Paton et al., 2006*).

The pattern of results we observed is consistent with functional imaging studies showing a role for the human OFC in signaling changes in food value specific to the food consumed (*Small et al., 2001*; *Gottfried et al., 2003*). The present data shed new light on value-based goal selection. Apparently, either during or soon after the value updating process, the new value is transmitted from area 13 to area 11 and potentially to other prefrontal cortical regions as well. The fact that area 13 does not need to be active during the choice phase reveals that area 13 is not an essential part of the circuitry generating goals for action.

Human fMRI studies suggest that the anterior OFC may represent foods at an abstract level; the posterior OFC in humans is more active in response to pictures of food relative to nonfood objects (*van der Laan et al., 2011*). Based on BOLD changes consequent to sensory-specific adaptation, *Klein-Flugge et al. (2013)* proposed that the anterior OFC, in the region of area 11, houses a stimulus–food outcome representation, whereas the more posterior OFC, in the region of area 13, houses representations of reward identity, independent of associated stimuli. This pattern of results is consistent with our findings, but leaves unanswered how visual sensory input translates this knowledge into goals.

Because area 11 can mediate adaptive choices independently of area 13, we conclude that area 11 and associated circuits link visual inputs with the current, updated value of particular food outcomes. This could occur within area 11, or it could occur in concert with anatomically related areas such as the ventral lateral prefrontal cortex (VLPFC), area 12, a region known to be important for visual attentional selection (*Rushworth et al., 2005*) and for switching among competing behavior-guiding rules (*Bussey et al., 2001*; *Rossi et al., 2007*; *Baxter et al., 2009*; *Buckley et al., 2009*).

*West et al. (2011)* have performed an experiment like the present one and found that bilateral inactivation of the OFC disrupted both value updating and goal selection based on those valuations. Because they infused muscimol into a central location within the OFC, their infusions probably affected both areas 11 and 13 simultaneously, which accounts for the difference between their results and ours. Other differences, such as the species tested (pig-tailed vs. rhesus macaques), the type of stimulus material (images vs. objects), and the pharmacological agent used (muscimol vs. THIP) seem unlikely accounts, although further work is necessary to rule out these variables.

## Feedforward and feedback projections within the OFC

Anatomical studies have suggested a hierarchical organization in the anteroposterior dimension of the OFC. Specifically, by analogy with sensory corticocortical connections, the posteriorly-directed projections within the OFC are said to follow a "feedforward" projection pattern characterized by projections that arise in supragranular layers (layers 2–3) of the cortex (*Carmichael and Price, 1996*). For example, the preponderance of cells giving rise to the posteriorly-directed projections from area 11 to area 13 and, likewise, from area 13 to the agranular insular cortex, arise from layers 2 and 3. This picture is consistent with a model in which visual sensory representations in area 11 subsequently activate representations in more posterior areas, where they become integrated with updated value signals. When area 13 is inactive during the probe tests, however, other regions, such as area 12 of the VLPFC or the anterior insula, are sufficient to provide the information about updated value.

## From goal selection to action

To influence behavior, the OFC must eventually affect the motor system. There are at least two models of how value-guided choice might be implemented. One idea posits a serial model (*Padoa-Schioppa, 2011*) in which offer values are compared, perhaps in a common currency. The offer with the greater value is selected, and then an action is selected to implement that choice. Another model suggests that value signals could influence processing in visuomotor pathways that plan movements through a mechanism akin to top–down attention and biased competition (*Pastor-Bernier and Cisek, 2011*). By either model, these influences of current value on choice might be mediated by connections between the OFC and the ventral, medial, and dorsolateral prefrontal cortex (*Barbas and Pandya, 1989*; *Carmichael and Price, 1996*). The OFC and the VLPFC contribute to multisynaptic pathways to dorsal premotor areas, mainly via dorsal and dorsolateral prefrontal

cortex (*Takahara et al., 2012*), and cingulate premotor areas, mainly via medial prefrontal cortex (*Morecraft and Van Hoesen, 1998*; *Morecraft et al., 2012*). The VLPFC in particular receives strong projections from areas 13 and 11 as well as from the basolateral amygdala (*Porrino et al., 1981*; *Saleem et al., 2014*). Thus, the OFC and the adjacent VLPFC are well situated to influence the selection of targets for action.

## Conclusions

Taken together, the results from inactivating areas 11 and 13 selectively demonstrate specialized functions for these two components of the macaque OFC. The posterior component, area 13, functions in conjunction with the basolateral amygdala to update the valuation of expected reward outcomes, based on an animal's current satiation state. The anterior component, area 11, plays a critical role in translating this knowledge into goals that produce an advantageous outcome. Importantly, area 11 is needed for choosing which of two visual stimuli to choose when both stimuli are abstractly associated with a particular reward outcome, but not for choices between two visible foods. The inability to translate abstract valuation knowledge into advantageous choices resembles the goal neglect that occurs after damage to the frontal lobe in humans.

# Materials and methods

## Monkeys

Six male rhesus monkeys (*Macaca mulatta*), experimentally naïve at the beginning of training, served as subjects. Five of the monkeys participated in the main experiment, and one additional monkey served as a subject in a subset of the control procedures only. All monkeys were housed in rooms kept on a 12-hr light/dark cycle (lights on at 7:00AM) and testing occurred during the light period. Four of the six monkeys were housed individually while the remaining two were pair housed with monkeys participating in a different experiment. At the beginning of the study, the monkeys ranged in weight from 6.4 to 9.5 kg. For the duration of the study, the monkeys were given controlled access to food to ensure sufficient motivation to respond in the test apparatus. Water was available ad libitum in the home cage. All procedures were reviewed and approved by the NIMH Animal Care and Use Committee.

## Apparatus and materials

Testing was carried out in an automated apparatus consisting of a microprocessor linked to a 15-inch color monitor fitted with a touch-sensitive screen. Rewards consisted of two of the following three foods: M&Ms (Mars Candies, Hackettstown, NJ), peanuts, and Skittles (Mars Candies, Hackettstown, NJ). Rewards were dispensed from automated dispensers (Med Associates, St Albans, VA) mounted on top of the test chamber. All tasks were controlled and behavioral data collected by a computer using custom software (Ryklin Software Inc., New York, NY; software is available for download: ftp://helix.nih.gov/lsn/beethoven/Operant_Testing_System.exe).

During each test session, the monkey was seated in a primate chair inside a light- and sound-attenuating test chamber. A fan mounted in the ceiling of the chamber provided ventilation and masked extraneous noise. Visual stimuli consisted of rectangular clipart images, approximately 55 mm x 55 mm. The monkey's head was approximately 230 mm from the monitor screen.

## Pretraining

Monkeys were first trained to touch a monitor screen using standard shaping procedures (*Murray et al., 1993*). They then learned a standard single-pair, visual discrimination. One image of the pair was arbitrarily designated as the rewarded image (S+), and the other was designated as the nonrewarded image (S−). The monkey was required to touch one of the two images on the screen. The selected image was then shown with a red frame and flashing red dot in the center. The nonselected image disappeared. If the selected image was the S+, touching it again led to delivery of a ½ peanut reward and visual feedback of the image on the screen for an additional 1 s. If the selected image was the S−, then touching it again resulted in the screen going blank. The intertrial interval (ITI) was set at 5 s after a correct (rewarded) response, and 10 s after an incorrect (nonrewarded) response. Criterion was set at a mean of 90 percent correct responses over three consecutive 60-trial

sessions. Finally, monkeys were required to discriminate 20 image pairs, repeated 3 times each per session. Stimuli were novel at the beginning of training, and general methods were the same as for the single-pair discrimination. Criterion was set at a mean of 90 percent correct responses over five consecutive sessions.

## Reinforcer devaluation task

After completing pretraining, each monkey began training on the devaluation task. Training began with the acquisition of a list of 60 fixed image pairs presented for visual discrimination (*Figure 1a*). All images were novel at the beginning of the experiment. One image of each pair was arbitrarily designated the S+ and the other the S−. On each trial, the two images of a pair were presented simultaneously, 150 mm apart, to the left and right of the screen center. If the monkey touched the S+, the S− image disappeared, and the selected image remained on the screen for one additional second while a food reward was delivered. If the monkey touched the S−, the screen went blank. There was no correction for errors. Each image pair was displayed once per session, in random order, yielding a total of 60 trials. The location of the S+ (left or right) also followed a pseudorandom Gellermann order. Each monkey had a fixed image pair list that was unique to the animal. For 30 of the image pairs, food 1 (e.g., ½ peanut) was delivered when the S+ was selected. For the remaining 30 pairs, food 2 (e.g., an M&M) was delivered when the S+ was selected. The S+ image-food assignments were fixed throughout training. The ITI was 5 s after correct responses and 10 s after incorrect responses. Criterion was set at a mean of 90 percent correct responses over five consecutive sessions (i.e., 270 or more correct responses in 300 trials).

After monkeys had reached criterion on the 60 pairs, but before data collection began in the main task, monkeys were transitioned to a four day per week test schedule (*Figure 3*) adapted from *Wellman et al. (2005)*. Day 1 was a review of the 60 discrimination problems, presented just as in initial learning. Day 2 comprised a probe test to assess monkey's baseline choices. S+ images assigned to different foods (food 1 and food 2) were pitted against each other, yielding 30 trials per session. Whichever image was chosen, the appropriate food was delivered. The images were paired anew for each probe test, with the constraint that pairs of images presented for choice always comprised one image associated with each of the two foods. Day 3 repeated the review of the 60 pairs, as on Day 1. The session on Day 4 followed the same procedure as on Day 2, but was carried out after selective satiation. In addition, this was the day on which we carried out infusions of THIP or saline, either before or after the selective satiation procedure.

The measure of interest was the number of food-1 and food-2 associated images selected on Day 4 (after selection satiation) relative to Day 2 (baseline), which reflects each monkey's ability to adaptively shift away from choosing images paired with the devalued food. We calculated proportion shifted according to Eq. 1, where F1 and F2 represent choices of the images paired with the two food types on weeks in which that food type was devalued, and subscripts D and N respectively represent the Devaluation day (Day 4) and the day of that same week on which that food was Not devaluated (Day 2). Thus, proportion shifted is the total number of image choices shifted due to devaluation, as a fraction of the total possible shift:

$$Proportion\ shifted = \frac{(F1_N - F1_D) + (F2_N - F2_D)}{F1_N + F2_N}$$

## Selective satiation procedure

The selective satiation procedure has been described in detail elsewhere (*Izquierdo et al., 2004*). Unlike earlier studies, where food was given in the home cage, we delivered the to-be-sated food while the monkeys remained seated in a primate chair. This was done to maintain strict control of the elapsed time between satiety and infusions, and between the infusions and initiation of test sessions.

## Food choices after selective satiation

To determine whether satiety mechanisms were intact, monkeys were given additional probe tests. In this series of tests, instead of being evaluated for image choices, monkeys were evaluated for food choices (*Figure 3*, Control manipulations). Each monkey sat in the testing room with the

experimenter. After undergoing the selective satiation procedure, the monkey was given a series of forced-choice trials between food 1 and food 2, presented on a two-well test tray. The placement of the foods (left and right) was pseudorandomized using a Gellermann schedule. To complete a trial, the monkey selected one of the two foods. The experimenter recorded the choice. Each session comprised 30 trials separated by 10 s.

## Surgery

All six monkeys underwent three stages of surgery under general anesthesia. In the first operation, monkeys were fitted with a titanium head post held in place with self-tapping titanium screws. After a minimum of four weeks, each animal received a second operation to implant an infusion chamber (Section on Instrumentation Core, NIH, Bethesda, MD). The chamber—which had interior dimensions of 26.5 mm × 46 mm and was fabricated from Ultem plastic—was fixed to the cranium using dental acrylic; ceramic screws positioned around the outside of the chamber served to anchor the implant. In the third and final stage of surgery, two craniotomies were made, one per hemisphere, within the bounds of the chamber. In a few instances, the bone regrew over the target areas and an additional operation was required to re-open the craniotomy.

During surgery, aseptic procedures were used. Anesthesia was induced with ketamine hydrochloride (10 mg/kg, i.m.) and maintained with isoflurane (1.0–3.0%, to effect). Heart rate, respiration rate, blood pressure, expired $CO_2$, and body temperature were monitored during surgery, and isotonic fluids were given throughout. Cefazolin antibiotic (15 mg/kg, i.m.) was given to prevent infection. The drug was administered for one day before surgery and for one week after surgery. Monkeys received the analgesic ketoprofen (10–15 mg, i.m.) at the end of surgery and for two additional days postoperatively. This was followed by 100 mg of ibuprofen for the next 5 days.

## MRI scans

Each monkey received multiple structural MRI scans, as needed, to guide chamber placement, cannula placement for infusions, and to confirm that infusions reached the intended targets. In general, the first scan, which was used to guide chamber placement, was carried out in a horizontal-bore 1.5 or 3 T scanner. The monkey was sedated with a mixture of ketamine (15 mg/kg, i.m.) and medetomidine (20 µg/kg, i.m.), supplemented as needed. Glycopyrrolate was administered to reduce secretions (0.01 mg/kg, i.m.) and ketoprofen (10–15 mg, i.m.) was given as an analgesic. Monkeys were placed in a MR-compatible stereotaxic frame for the duration of the scan.

To guide cannula placement and to calculate the depth of infusion cannulae, the monkeys received an additional structural scan using a 4.7T vertical-bore scanner. For this scan, the chamber was filled with a gadolinium–saline solution (1:1200 v:v; gadolinium, Bayer HealthCare, Wayne, NJ) and the chamber grid was placed in the chamber to allow visualization of the vertical channels through which infusion cannulae would be inserted (see Infusions). Finally, to verify that our infusions reached the intended target, each monkey received a gadolinium–saline infusion (1:100 v:v, 2.0 µl/ site) targeted just dorsal to the tissue of interest, and then received a scan in the vertical-bore magnet as described above. This final type of scan was conducted once for area 11 and again for area 13 before data collection for that area commenced. Sedation was achieved using a mixture of ketamine (5–20 mg, i.m.) and diazepam (0.5–1.5 mg/kg, i.m.), supplemented as needed. Glycopyrrolate was administered to reduce secretions (0.015 mg/kg, i.m.).

## Infusions

We carried out infusions in area 13 and area 11, one area followed by the other. The order in which areas were studied was balanced across subjects. The experimental manipulations were carried out on Day 4 (*Figure 3*). Infused compounds were either saline (vehicle; phosphate buffered saline) or the drug THIP (18 mM; Tocris, Bristol, UK) which, like muscimol, is a $GABA_A$ agonist. In each case, the drug was freshly dissolved in vehicle, and the solution, pH 7.0–7.5, was sterile filtered (Corning, Corning, NY) before injection.

Prior to each infusion, the chamber cap was removed and a sterile grid was inserted into the chamber. The grid fit snugly inside the chamber and contained vertical channels spaced 1 mm apart in the form of two 19 x 19 arrays, one each over the left and right hemisphere (*Talbot et al., 2011*). The two-dimensional coordinate frame of the grid permitted reliable placement of the infusion

cannulae. By using the same x, y, and z coordinates from one infusion to the next, the target location could be reproduced. When the grid was in place, sterile guide tubes (thin walled 24 gauge, Component Supply Company, Fort Meade, FL) and infusion cannulae (regular walled 30 gauge, Component Supply Company, Fort Meade, FL) were inserted into the brain. The infusions were performed while the monkeys were awake and seated in the primate chair, with their heads restrained with the implanted head posts.

Infusions were conducted using a Harvard Apparatus infusion pump (PHD 2000; Harvard Apparatus, Holliston, MA) and 100 µl Hamilton syringes (model 1710TLL; Hamilton Syringe Company, Reno, NV). Each infusion was administered with a flow rate between 0.18–0.25 µl/min over the course of 10–12 min, for a total volume between 1.98–2.34 µl. The vast majority of infusions used a total volume of 2.0 µl with a flow rate of 0.18 µl/min. Thus, drug or saline was infused over a period of ~11 min. The infusion period was immediately followed by a 10–15 min wait with the pump turned off but the infusion cannulae and guide tubes in place to promote diffusion at the site of infusion and to limit drug traveling up the track upon withdrawal of the needle. When the wait-period was completed, the guide tubes and infusion cannulae were withdrawn, the grid was removed, the chamber was rinsed with sterile saline, and the chamber cap was replaced. Approximately 20 min elapsed between the end of the infusion and the beginning of the test session or satiation procedure.

In each region of the OFC investigated, two THIP infusions and two saline infusions were carried out *before* the selective satiation procedure, one for each food. Similarly, two THIP infusions and two saline infusions were carried out *after* the selective satiation procedure, again, one for each food.

## Control infusions

In addition to the saline (vehicle) infusions conducted as part of the main experiment, we carried out other infusions as needed to address interpretational issues. Intermixed with the experimental conditions, an additional THIP infusion was given without satiation. For this session, the monkey waited in the chair for the time it would take to complete the selective satiation procedure. This was done to test the possibility that either the delay period or the presence of the drug in the brain tissue alone was enough to significantly alter performance (*Figure 3*, Control manipulations, no satiation).

In area 11 only, an additional infusion with THIP was given in four of the five monkeys. This infusion was performed prior to testing on Day 3 of the test schedule, on which the animals were presented with the list of 60 pairs, to determine if it disrupted the animals' performance.

Two of the monkeys from the main experiment and a third monkey, used for this procedure only, underwent additional infusions to test for effects of the infusions on satiety mechanisms. In each monkey, we carried out four drug and two saline infusions before and after selective satiation involving either food 1 or food 2.

## Reconstruction of infusion sites

For each monkey, the structural MR scan images were matched to drawings of coronal sections of a standard rhesus monkey brain at 1 mm intervals through the entire frontal lobe. Using the structural scans acquired with gadolinium-saline solutions in the chamber grid, an imaginary line was extended from the grid hole used to target each area (area 13 or area 11) to the orbital surface. This site on the surface was then plotted onto the standard section and then transferred to the ventral view.

## Tracer injections

At the conclusion of the experiment, the retrograde tracers cholera toxin subunit B (CTB; List Biological, Campbell, CA; 1–2%) and Fast Blue (Sigma-Aldrich, St. Louis, MO; 3%) were injected into area 11 and into area 13, respectively, in one monkey. The reverse was done in a second monkey; CTB was injected into area 13 and Fast Blue into area 11. Both sets of tracer injections were performed unilaterally. Hamilton syringes were inserted into the chamber grid at the same x, y location and to the same depth as for the infusions. After a survival period of 13 days, the monkeys were deeply anesthetized with Euthanasia (0.1 ml/kg, i.v.) and perfused transcardially with normal saline, followed by 4% paraformaldehyde in 0.1M phosphate buffer. The brain was extracted from the skull, blocked, and cryoprotected through a series of glycerols. After 3–4 days, the brain was frozen in dry ice and isopentane, and cut in the coronal plane at 40 µm-thick sections on a sliding microtome. One to two

parallel series of sections were immediately mounted on gelatin-coated slides, air-dried, and cover-slipped with DPX (Sigma-Aldrich) for the examination of fluorescent tracer Fast Blue. The other series was processed immunohistochemically with the Avidin/Biotin immunoperoxidase method for CTB labeling.

## CTB immunostaining

To visualize CTB, sections were first rinsed in 0.1M phosphate buffered saline (PBS, pH 7.4), and then incubated for 2 hr in blocking serum consisting of 0.3% Triton X-100 (TX-100), 2% bovine serum albumin (BSA), 0.3% hydrogen peroxidase ($H_2O_2$), and 15% normal rabbit serum (NRS) in PBS. Tissue was then incubated in the primary antibody solution (1:5000 anti-CTB, added to the serum solution consisting of 2% BSA, 5% NRS, and 0.2% TX-100 in PBS) for 65 hr at 4°C. After several washes in PBS, sections were then incubated in the secondary antibody solution (1:200 anti-goat IgG, added to the same serum solution as described in the previous step) for 1.5 hr at room temperature, followed by another wash in PBS. The sections were then processed with the avidin/biotin staining kit (Vector ABC Elite) for 90 min at room temperature, after which sections were washed in PBS and placed in a 0.035% DAB solution (3,3-diaminobenzidine tetrahydrochloride as chromogen; Sigma #D5637). After 10 min, approximately 0.0125% hydrogen peroxide was added to initiate the staining reaction. The DAB reaction was stopped when satisfactory contrast was achieved (usually 2–5 min). After a final rinse in phosphate buffer, sections were mounted on gelatin-coated slides, air-dried, and dehydrated through ascending grades of ethanol concentrations before being cleared in xylenes and coverslipped in DPX (Sigma-Aldrich, St. Louis, MO).

## Acknowledgements

We thank Ping-yu Chen for technical support. Structural MRI scans were carried out in the Neuro-physiology Imaging Facility Core (NIMH, NINDS, NEI) and in the NMR Facility (NINDS). This work was supported by the Intramural Research Program of the National Institute of Mental Health.

## Additional information

### Funding

| Funder | Grant reference number | Author |
|---|---|---|
| National Institute of Mental Health | MH002887-10-LN | Elisabeth A Murray |

The funders had no role in study design, data collection and interpretation, or the decision to submit the work for publication.

### Author contributions

EAM, JT, Conception and design, Acquisition of data, Analysis and interpretation of data, Drafting or revising the article; EJM, KSS, BMB, Acquisition of data, Analysis and interpretation of data, Drafting or revising the article

### Author ORCIDs

Elisabeth A Murray, http://orcid.org/0000-0003-1450-1642

### Ethics

Animal experimentation: All research was carried out in strict adherence to the laws and regulations of the U.S. Animal Welfare Act (USDA, 1990) and Public Health Service Policies (PHS, 2002), as well as nongovernmental recommendations of the National Research Council as published in the ILAR 'Guide for the Care and Use of Laboratory Animals'. All procedures were reviewed and approved by the National Institute of Mental Health Animal Care and Use Committee.

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
