## [Decision Letter]

Thank you for submitting your work entitled "Specialized areas for value updating and goal selection in the primate orbitofrontal cortex" for consideration by *eLife*. Your article has been reviewed by Timothy Behrens (Senior Editor) and three reviewers, one of whom is a member of our Board of Reviewing Editors.

The reviewers have discussed the reviews with one another and the Reviewing editor has drafted this decision to help you prepare a revised submission.

Summary:

This study used reversible, pharmacological inactivation (injection of a GABA agonist, THIP) to identify dissociable roles of parts of the primate orbitofrontal cortex (OFC areas 11, which is more anterior, and 13, which is more posterior) on reinforcer devaluation in monkeys. The basic test was to inject THIP either before or after a satiation procedure (giving the monkey a lot of one kind of treat) and test whether the monkey then altered the proportion of times choosing that treat versus another. They used a very nice within-subject experimental design in which each monkey was tested under a variety of conditions, including infusion of saline or THIP, injection into area 11 or 13, and injection before or after satiation. They found a very clean dissociation between the two regions. Inactivating area 13 prior to the selective satiation procedure impaired the ability of the animal to change its behavior, but if the inactivation occurred later (prior to the choice trials) the animals' performance was normal. In contrast, inactivating area 11 had no effect when it occurred prior to satiation, but disrupted performance if it was inactivated prior to the probe tests. Along with a number of important controls, the results provide strong evidence for a functional dissociation between the two areas: area 13 appears to be responsible for updating the value of stimulus-outcome associations, whereas area 11 appears to be responsible for using those updated values to guide choice.

All three reviewers agree that this is an important paper for determining OFC function. To our knowledge, it is the first study to demonstrate a double dissociation between anterior and posterior OFC. Furthermore, it uses the primate model, where OFC is readily comparable to human OFC. The experiments are well designed and include controls that help to eliminate more general changes in preference behavior, discrimination ability and response to satiation. The paper is also very clearly written.

Essential revisions:

1) It would be instructive to show data from individual monkeys instead of just group averages with error bars, to make it easier to assess the consistency of the results.

2) A fuller discussion of the effects of area 11 inactivation would be useful. The authors state early in the Results that "infusions administered before the selective satiation procedure inactivated the OFC during the selective satiation procedure as well as during the choice trials of the probe test. Thus, disruption of devaluation effects by THIP infusions before selective satiation would indicate a failure of either value updating or goal selection." Accordingly, why was there no effect of area 11 inactivation before the satiation procedure, which according to the previous quote (reflecting the duration of THIP effects) should still lead to a behavioral effect if applied to a region involved in goal selection?

---

## [Author Response]

*Essential revisions:*

*1) It would be instructive to show data from individual monkeys instead of just group averages with error bars, to make it easier to assess the consistency of the results.*

We have revised Figure 4 to show data from individual subjects. This is commonly done in neuropsychological investigations with a small number of subjects, so we were happy to do so.

*2) A fuller discussion of the effects of area 11 inactivation would be useful. The authors state early in Results that "infusions administered before the selective satiation procedure inactivated the OFC during the selective satiation procedure as well as during the choice trials of the probe test. Thus, disruption of devaluation effects by THIP infusions before selective satiation would indicate a failure of either value updating or goal selection." Accordingly, why was there no effect of area 11 inactivation before the satiation procedure, which according to the previous quote (reflecting the duration of THIP effects) should still lead to a behavioral effect if applied to a region involved in goal selection?*

We have inserted a new paragraph in the Discussion section to explain this phenomenon. The new text reads:

“An additional aspect of our double dissociation deserves comment. Inactivation of area 11 before the selective satiation procedure, which had no effect on behavior, would have persisted into the choice phase, when THIP infusions into area 11 did cause an impairment. But it is reasonable to expect an impairment to follow choice-phase inactivation of area 11, regardless of how long its neurons had been inactivated. We do not believe that the effect of pre-satiation THIP infusions could have dissipated in time for area 11 to participate in goal selection. Infusion of similar volumes of THIP into the pulvinar produced behavioral effects in monkeys lasting several hours (Wilke et al., 2013), and smaller volumes administered in rat superior colliculus affected behavior for at least two hours (Di Scala et al., 1983), both well beyond the elapsed time between pre-satiation infusion and choice tests in the present study.

Instead, our negative results after pre-satiation infusions probably reflect the cooperative function of other prefrontal areas, which would also explain the relatively mild effect of choice- phase area 11 inactivations. The impairment when area 11 was inactive only during goal selection, but not when it was inactive during bothvalue updating and goal selection, suggests that area 11 plays a dominant role in a partially redundant goal-selection network. Another possibility is that when area 11 was inactivated before the selective satiation procedure, it was effectively taken ‘offline’, rendering it unable to receive updated valuation signals and therefore unable to participate in goal selection later. According to this account, when area 11 had received updated valuation signals during the satiation procedure it made a significant contribution to goal selection, but otherwise did not. Presumably, neighboring prefrontal areas could mediate goal selection in this circumstance.”